# MITF Downregulation Induces Death in Human Mast Cell Leukemia Cells and Impairs IgE-Dependent Degranulation

**DOI:** 10.3390/ijms24043515

**Published:** 2023-02-09

**Authors:** Elizabeth Proaño-Pérez, Laia Ollé, Yanru Guo, Cristina Aparicio, Mario Guerrero, Rosa Muñoz-Cano, Margarita Martin

**Affiliations:** 1Biochemistry and Molecular Biology Unit, Biomedicine Department, Faculty of Medicine and Health Sciences, University of Barcelona, 08036 Barcelona, Spain; 2Clinical and Experimental Respiratory Immunoallergy (IRCE), Institut d’Investigacions Biomediques August Pi i Sunyer (IDIBAPS), 08036 Barcelona, Spain; 3Faculty of Health Sciences, Technical University of Ambato, Ambato 180105, Ecuador; 4Allergy Department, Hospital Clinic, University of Barcelona, 08036 Barcelona, Spain; 5RICORS, Instituto de Salud Carlos III, 28220 Madrid, Spain

**Keywords:** MITF, miRNA, cell survival, mast cells, degranulation, mastocytosis, mast-cell-derived diseases

## Abstract

Activating mutations in KIT (CD117) have been associated with several diseases, including gastrointestinal stromal tumors and mastocytosis. Rapidly progressing pathologies or drug resistance highlight the need for alternative treatment strategies. Previously, we reported that the adaptor molecule SH3 binding protein 2 (SH3BP2 or 3BP2) regulates KIT expression at the transcriptional level and microphthalmia-associated transcription factor (MITF) expression at the post-transcriptional level in human mast cells and gastrointestinal stromal tumor (GIST) cell lines. Lately, we have found that the SH3BP2 pathway regulates MITF through miR-1246 and miR-5100 in GIST. In this study, miR-1246 and miR-5100 were validated by qPCR in the SH3BP2-silenced human mast cell leukemia cell line (HMC-1). MiRNA overexpression reduces MITF and MITF-dependent target expression in HMC-1. The same pattern was observed after MITF silencing. In addition, MITF inhibitor ML329 treatment reduces MITF expression and affects the viability and cell cycle progression in HMC-1. We also examine whether MITF downregulation affected IgE-dependent mast cell degranulation. MiRNA overexpression, MITF silencing, and ML329 treatment reduced IgE-dependent degranulation in LAD2- and CD34^+^-derived mast cells. These findings suggest MITF may be a potential therapeutic target for allergic reactions and deregulated KIT mast-cell-mediated disorders.

## 1. Introduction

Mastocytosis is a myeloproliferative neoplasm characterized by mast cell (MC) proliferation in various tissues, including skin, bone marrow, gastrointestinal tract, liver, spleen, or lymph nodes [1]. Hematologic disorders are associated with increased mast cell proliferation. Clinical features include flushing, pruritus, abdominal pain, diarrhea, hypotension, syncope, and musculoskeletal pain. Symptoms exhibited in mastocytosis result from the increase of MC mediators’ release and MC infiltration into target organs [2,3]. Activating mutations of KIT, a receptor tyrosine kinase, are central to mastocytosis pathogenesis, enabling the proliferation and survival of abnormal mast cells in affected tissues [4]. Asp816Val (D816V) KIT mutation is the hallmark of mastocytosis, and it is located in the catalytic domain of the receptor, leading to mast cell proliferation and survival [2,5]. Mutations, including V560G, D816Y, D816F, D816H, and E839K, have been identified in mast cell lines, mast cell leukemia, and pediatric mastocytosis [6,7].

Previously, our group established a relationship between the KIT receptor de-adaptor molecule SH3BP2 and the microphthalmia-associated transcription factor (MITF) in mast cells [8] and gastrointestinal stromal tumors (GIST) [9]. We reported a reduction of KIT expression and MITF levels after silencing SH3BP2. The functional outcome was a reduction in cell survival and proliferation.

MITF belongs to the MiT family (TFEB, TFEC, and TFE3) and forms homodimers and heterodimers. MITF regulates gene expression by binding to E-box motifs (CA [C/T]GTG) or by establishing associations with and modulating the activity of other transcription factors [10]. MITF regulates the expression of genes critical for the survival of melanocytes and melanoma cells, such as the anti-apoptotic proteins BCL2 and BCL2A1 and the cell cycle regulator CDK2 [11,12,13]. Moreover, oncogenic MITF amplifications are found in melanomas, and overactivity of MITF drives melanoma formation [14]. In addition to melanocytes, MITF is crucial in osteoclast and mast cell differentiation [15]. Regarding mast cells, MITF is downstream of KIT and FcεRI pathways [16,17,18]. An increased MITF activity in mast cells has recently been associated with anaphylaxis [19].

To obtain insights into the KIT-SH3BP2-MITF pathway, a miRNA microarray was performed in SH3BP2-silenced GIST882 and GIST48 cell lines (imatinib-sensitive and resistant cells) compared with non-silenced cells [20]. This microarray showed a different miRNA pattern when SH3BP2 was silenced. The most up-regulated miRNAs were validated as predictive of MITF expression, and miR-1246 and miR-5100 were identified in GIST [20].

In this study, we investigated miRNAs regulated by SH3BP2 silencing, the role of MITF activity, and the efficacy of MITF pathway inhibitor, ML329 [21], in inhibiting the survival and cell proliferation in HMC-1 as a cellular model for mastocytosis. Moreover, we analyzed the ability of MITF levels in IgE-dependent mast cell degranulation.

## 2. Results

### 2.1. Analysis of Up-Regulated Mirnas That Target MITF Validation in Hmc-1 Cell Lines

This study focuses on microRNAs up-regulated in SH3BP2-silenced GIST cells targeting MITF [20]. For that purpose, we used TargetScan [22], miRtar [23], miRwalk 2.0 [24], microT CDS [25], and mirDIP [26] to predict which of the 21 most up-regulated miRNAs found in the microarray carried out by Exiqon could target MITF. We validated two miRNAs: miR-1246 and miR-5100, targeting MITF in GIST cell lines [20]. To validate these miRNAs in HMC-1, a quantitative real-time PCR was carried out in SH3BP2-silenced HMC-1 cells (Figure 1A). Both were expressed significantly after SH3BP2-silencing (Figure 1).

### 2.2. MiR1246 and miR5100 Target MITF, and Overexpression Significantly Affects Cell Proliferation and Cell Cycle Progression

These miRNAs putatively bind to MITF, so we overexpressed them in the HMC-1 cell line to check MITF protein levels. Figure 2 shows that overexpression of miR-1246 and miR-5100 efficiently causes the downregulation of MITF protein levels. These results corroborate that these miRNAs target MITF and regulate their expression post-transcriptionally. Moreover, MITF-dependent targets, such as BCL2 [12] and CDK2 [13], are decreased (Figure 2A). A KIT and BCL2 reduction diminished cell proliferation in HMC-1 cells (Figure 2B). The CDK2 decline prompted us to analyze the cell cycle, and an arrest in the G2/M phase was detected (Figure 2C). CDK2 has been described to regulate both G1/S and G2/M transitions [27]. Altogether, these results indicate that these miRNAs may regulate MITF-dependent survival and cell cycle progression. GFP-miRNAs expression is shown in Appendix A.

### 2.3. MITF Silencing Reduces KIT, BCL2, and CDK2 Expression in HMC-1 Cell Lines

To analyze the role of MITF in HMC-1, MITF was silenced by lentivirus containing shRNAs sequences. Among the three different sequences tested—shRNA MITF-1, shRNA MITF-2, and shRNA MITF-3—sequences two and three were effective on day five (Appendix A). Expression of MITF and MITF-dependent targets KIT, BCL2, and CDK2 were analyzed. Our results show that MITF silencing was accompanied by a reduced BCL2, CDK2 expression, and KIT, to a lesser extent (Figure 3A, Appendix A). Consistently, MITF silencing decreased cell viability (Figure 3B) and cell cycle arrest in the G2/M (Figure 3C). These data reinforce the above data using miRNAs that target MITF (miRNA1246, miRNA5100) in HMC-1 (Figure 2) and are following MITF’s role in cell survival and cycle progression analyzed in GIST cell lines [20].

### 2.4. MITF Activity Inhibition by ML329 Reduces Cell Proliferation and Cycle Arrest

MITF pathway inhibitor ML329 has been reported to inhibit cell survival and proliferation in melanoma [21,28]. Next, we analyzed ML329 action in HMC-1. Remarkably, ML329 treatment impairs KIT, BCL2, and CDK2 levels in HMC-1 (Figure 4A), alters proliferation (Figure 4B), and impairs the cell cycle, showing a significant arrest in the S phase (Figure 4C).

### 2.5. MITF Downregulation Promotes Apoptosis in HMC-1 Cells

Next, we assayed whether MITF downregulation leads to cellular apoptosis. For that purpose, caspase 3/7 activity was measured on overexpressed miRNAs: miR-1246 and miR-5100 in HMC-1 cells (Figure 5A), MITF-silenced HMC-1 cells (Figure 5B), and ML329-treated cells (Figure 5C). Our results show increased caspase 3/7 activity in all cases where the MITF level was reduced.

### 2.6. MITF Downregulation Reduces IgE-Dependent Mast Cell Degranulation

We were also interested in analyzing whether a reduction in MITF levels is consistent with a decrease in IgE-dependent mast cell degranulation. HMC-1 is not a good model for studying IgE-dependent degranulation since it lacks surface expression of the high-affinity IgE receptor [29]. Thus, we analyzed MITF downregulation in LAD2 cells and CD34^+^-derived mast cells. CD63 staining was used to measure degranulation, and the analysis was performed in the live cell population (propidium iodide negative). As shown in Figure 6, MITF downregulation by miRNAs 1246 and 5100 and ML329 treatment significantly reduces antigen-dependent mast cell degranulation. GFP-miRNAS overexpression was monitored by fluorescence microscopy (Appendix A).

## 3. Discussion

Mastocytosis is a heterogeneous disease associated with the clonal proliferation of mast cells (MC) driven commonly by the oncogenic KIT mutation (D816V). Tissues and organs affected are skin, bone marrow, gastrointestinal tract, lymph nodes, liver, and spleen [30]. Treatment of the disease involves tyrosine kinase inhibitors; however, the D816V mutation offers resistance to a wide range of inhibitors [1,31,32]. The recently approved avapritinib highly specific oral inhibitor of KIT activation loop mutants, including D816V, has shown encouraging results [33,34]. Nevertheless, mastocytosis’s clinical and molecular heterogeneity requires studies beyond KIT approaches. Recently, the inhibitory receptor Siglec 7 was reported to inhibit human mast cell leukemia cells in vitro and in vivo growth [35]. In this study, we explored microphthalmia-associated transcription factor (MITF) and dependent targets in the context of mast cells and mastocytosis. Because of the limited availability of mastocytosis patients and the reduced MCs numbers that can be obtained from the patient’s bone marrow, we used HMC-1 cells, a mast cell line harboring D816V as a cellular model of mastocytosis, to characterize MITF’s role further. LAD2 cells and CD34^+^-derived mast cells were used to assess IgE-dependent degranulation in MITF downregulation.

The microphthalmia-associated transcription factor (MITF) is highly expressed in bone marrow biopsies from 9 of 10 patients with systemic mastocytosis and activating KIT mutations [17]. MITF expression is regulated by KIT-dependent signals and is required for the transformed phenotype of mastocytosis [36]. Normal KIT signaling and oncogenic signaling can regulate the expression of MITF at post-transcriptional levels via miR-539 and miR-381 [17]. KIT–MITF regulation is reciprocal since MITF regulates KIT expression by binding to its promoter in mast cells [37]. Previously, we reported that SH3BP2 silencing reduces KIT levels in human mast cells and HMC-1, leading to increased cell death. Interestingly, SH3BP2 silencing decreases MITF protein levels but not *MITF* mRNA levels [8], suggesting post-transcriptional regulation. Moreover, we observed the same results in gastrointestinal stromal tumor (GIST) cell lines where a gain of function mutations on KIT or PDGFRA were the oncogenic drivers. Thus, SH3BP2 silencing decreased oncogenic KIT/PDGFRA levels, reduced MITF levels, led to apoptosis in vitro, and decreased tumor growth in vivo [9]. Our recent data show that MITF is required for GIST cell survival, proliferation, and tumor growth in xenograft experiments [38]. Moreover, we have found that SH3BP2 silencing increases microRNAs (miR1246 and miR5100) that target MITF, reducing cell viability, proliferation, and cell cycle progression and enhancing caspase 3/7 activity [20]. In this study, miR-1246 and miR-5100 were up-regulated after SH3BP2 silencing in HMC-1. Overexpression of both miRNAs targets MITF and MITF-dependent targets. MiR-1246 and miR-5100 have been reported to be involved in apoptosis induction and tumor cell growth inhibition in several oncogenic processes [39,40,41,42,43]. However, other studies have also noted a pro-oncogenic role of these miRNAs [44,45,46,47]. These controversial results may be related to the differential expression of miRNAs in specific tissues and/or distinct networking partners [47]. Nevertheless, our data is consistent in two different KIT oncogenic cell models: GIST cell lines [20] and now HMC-1, suggesting that these miRNAs’ role in oncogenic-KIT-derived diseases may be apoptotic. MITF silencing reproduces the apoptotic phenotype with MITF-dependent target reduction, such as BCL2 and CDK2, and KIT, to a minor extent. KIT, CDK2, and BCL2 levels were more reduced after ML329 treatment than after miRNAs or shRNAs treatment. CDK2 has been described to control both G1/S and G2/M transitions [27]. Higher expression of double-negative CDK2 arrests cells in the mid-S phase, whereas lower expression progresses well through an early and mid-S phase, however arrests in late S/G2 [27]. Although MITF silencing has been reported not to affect KIT levels [17], we observed a sustained reduction of KIT when MITF levels are down. Interestingly, the MITF mutant mouse is strikingly similar to SCF or KIT-deficient mice regarding mast cells and melanocytes [15,48]. MITF downregulation is also obtained after ML329, an inhibitor of the MITF pathway in melanoma [21] treatment, reinforcing the whole data.

In addition to our results about MITF as an important pro-survival factor in HMC-1, MITF has a well-recognized role as a factor involved in the generation and function of mast cells [49]. MITF governs downstream signaling of the STAT5 required to differentiate pre-basophil and mast cell progenitors into mast cells [50]. MITF-A is the longest and most-expressed MITF isoform in human CD34^+^ progenitor-derived mast cells (hMCs) and HMC-1 cells [51].

Recently, the GATA2-MITF axis has been shown to be critical for IgE/MC-mediated anaphylaxis [52]. Indeed, GATA2 induces MITF expression and maintains the accessibility of MITF to the histidine decarboxylase (*HDC)* promoter. HDC is the enzyme responsible for histamine production from histidine. Furthermore, MITF increases the expression of PGD2 [53]. Altogether, MITF regulates the expression of critical proinflammatory mediators in mast cells. In this study, we show that MITF downregulation reduces degranulation. In that context, bone-marrow-derived MCs from *Mitf* -/- mice show hypogranularity and defective SCF-dependent migration that can be restored with *Mift* addition [54].

On the other hand, MITF activity can be repressed by the histidine triad protein (HINT1) in quiescent mast cells. Upon antigen-IgE crosslinking, MEK activation increases, leading to Lysine t-RNA synthetase (LysRS) phosphorylation (Ser 207 LysRS) and its translocation to the nucleus. LysRS produces diadenosine tetraphosphate (Ap4A) in the nuclear compartment, which binds to HINT1, dissociating the MITF/HINT complex and releasing MITF that can start target gene transcription [18]. A recent study highlighted the length of the phosphodiester linkage of Ap4A with the ability to dissociate the MITF/HINT complex [55]. Lately, our group described a mutation in the KARS gene that encodes LysRS associated with severe anaphylaxis to wasp venom [19]. We found that mutation P542R LysRS (proline replaced by arginine at amino acid 542) results in conformational changes and translocation of the mutant protein to the nucleus in quiescent mast cells. The structural analysis shows the mutant in an open state that resembles the phosphorylated serine 207 LysRS activated form. Interestingly, this mutation enhances MITF activity and increases the transcription of target genes, such as *Hdc* and *Cma*, in quiescent mast cells.

New forms of mastocytosis include patients without skin lesions and presenting with hymenoptera anaphylaxis [33]. Indeed, the association of clonal mast cell disorders with hymenoptera allergy seems to be more specific than that with food- or drug-induced systemic reactions. These data open the window to explore genes involved in mast cell activation in the context of mastocytosis.

MITF expression is conserved in primary and metastatic malignant melanomas and is imperative in distinguishing cutaneous melanomas from nonmelanocytic lesions [56,57]. Although raised from different cells, mastocytosis and melanoma share factors such as MITF, STAT3, and dependence on KIT signaling, suggesting some similarities in both pathologies [58,59]. Physiopathological implications connecting melanocytic tumors and abnormal proliferation of MCs have been reported [60], suggesting that patients with cutaneous mastocytosis undergo full-body screening for melanocytic lesions.

Apart from melanoma, subsequent studies have suggested that MITF has multiple effects on cancers such as hepatocellular carcinoma [61], pancreatic cancer [62], lung cancer [63], and papillary renal cell carcinoma [64,65]. As explained above, we have found high similarities in MITF role and MITF-dependent targets in the oncogenic KIT-disease models analyzed. This study suggests that target MITF expression may be a therapeutic strategy, and further studies about MITF and MITF-dependent targets would be highly desirable.

## 4. Materials and Methods

### 4.1. Antibodies and Reagents

Mouse anti-3BP2 (clone C5), mouse anti-KIT (clone Ab81), mouse anti-BCL2, and mouse anti-CDK2 were purchased from Santa Cruz Biotechnology, Inc (Santa Cruz, CA, USA). Anti-MITF (clone D5G7V) was obtained (was provided by) from Cell Signaling Technology, Inc (Danvers, MA, USA). Mouse anti-β-actin (clone AC-40) and mouse anti-tubulin (DM1A) were purchased from Sigma (St. Louis, MO, USA). Anti-mouse and anti-rabbit IgG peroxidase Abs were acquired and bought from DAKO (Carpinteria, CA, USA) and Biorad (Hercules, CA, USA). ML329 was from Axon Med Chem (Groningen, The Netherlands).

### 4.2. Cell Culture

The human mast cell line, HMC-1 (KIT mutation in G560V and D860V), was obtained from J. H. Butterfield (Mayo Clinic, Rochester, MN) and was grown in Iscove’s media supplemented with 10% heat-inactivated FBS, penicillin (100 U/mL), and streptomycin (100 μg/mL) [29]. The LAD2 human mast cell line was kindly provided by Drs. A. Kirshenbaum and DD. Metcalfe (National Institutes of Health, Bethesda, MD, USA), and was grown in StemPro-34 media (Life Technologies, Carlsbad, CA), supplemented with StemPro-34 nutrient and L-glutamine (2 mM), penicillin (100 U/mL) and streptomycin (100 μg/mL), and 100 ng/mL SCF (Amgen, Thousand Oaks, CA) [66]. Primary human mast cells (MCs) derived from CD34-positive peripheral blood cells were obtained from buffy coats in vitro for eight weeks in the presence of 100 ng/mL IL-6 and 100 ng/mL SCF, as previously described [67]. After eight weeks, culture purity was assessed by surface FcεRI and KIT expression. The HEK 293LTV cell line (Cell Biolabs Inc, San Diego, CA, USA) was used for lentivirus production. The mycoplasma test is performed routinely in all cell lines used.

### 4.3. RNA Extraction, Retrotranscription, and PCR Assays

Total RNA was extracted with a miRCURY RNA Isolation Kit (Exiqon, Vedbaek, Denmark) from NT control and SH3BP2 knockdown HMC-1 cells. cDNA was generated by reverse transcription using the miRCURY LNA RT Kit. Quantitative, real-time PCR for miRNA PCR assay was performed using the miRCURY SYBR Green PCR Kit and following miRCURY LNA miRNA PCR assay protocol on a LightCycler^®^ 480 Instrument II (LifeScience Roche Indianapolis, IN, USA). miR-30c-5p were used as housekeeping miRNA genes.

### 4.4. Lentiviral Transduction

Lentiviral particles to silence the SH3BP2 and MITF gene expression were previously described [9,38]. Lentiviral vectors containing microRNA sequences of interest (pLenti-III-miR-GFP, by ABM Inc. Richmond, BC, Canada) were used for miRNA overexpression in HMC-1 cell lines. PLenti-III-mir-GFP-blank was the plasmid used as a control. MiRNA sequences (hsa-miR1246 and hsa-miR5100) can be retrieved from miRTarBase and were used previously [20].

### 4.5. GFP-miRNA Overexpression

MiR-CTL, miR-1246, or miR-5100 were overexpressed by lentiviral transduction. HMC-1 or LAD2 cells were transduced in the presence of 8 μL/mL of polybrene (Santa Cruz, CA, USA), and puromycin selection (1 μg/mL) was carried out after one day from transduction. miRNA-GFP expression was corroborated in both cell lines on the fourth day by microscopy fluorescence (Leica AF600, Wetzlar, Germany).

### 4.6. Cell Viability and Proliferation

Cell viability and proliferation were evaluated using a colorimetric assay (WST-1 based) (Version 17 Cell Proliferation Reagent WST-1, Roche Diagnostics, Germany) on the second, fourth, and sixth days after lentiviral transduction. The assay was performed according to the manufacturer’s protocol.

### 4.7. Caspases 3/7 Activity Assay

Caspase activity was assayed using the Caspase-Glo™ 3/7 Assay (Promega, San Luis Obispo, CA, USA), according to the manufacturer’s instructions.

### 4.8. Western Blotting

Western blotting was carried out as described [9,68]. Transduced cells were lysed on the fourth day of post-lentiviral infection. The total protein concentrations were determined using the ProteinAssay Dye Bio-Rad Kit (Bio-Rad Laboratories, Inc. Hercules, CA, USA), according to the manufacturer’s recommendations. Electrophoresis and protein blotting was performed using NuPage^TM^ 4–12%Bis-Tris Gel, 1.5 mm * 15 w (Invitrogen, Waltham, MA, USA), and electrotransferred to polyvinylidene difluoride (PVDF) membranes (Millipore, Bedford, MA, USA). Blots were probed with the indicated antibodies. In all blots, proteins were visualized by enhanced chemiluminescence (WesternBright^TM^ ECL, Advansta, San Jose, CA, USA).

### 4.9. Cell Cycle Analysis by Flow Cytometry

HMC-1 cells were collected on the fourth and sixth days after transduction. The cells were fixed with 70% ethanol at 4 °C overnight. Cells were washed twice with cold PBS buffer and resuspended in propidium iodide staining solution buffer for 30 min, as described elsewhere [69]. Data were acquired in FACS Calibur and analyzed using model Dean/Jet/Fox FlowJo 7.6 software.

### 4.10. Degranulation Assays

For IgE-dependent activation, cells were sensitized overnight with 0.1 mg/mL biotinylated human IgE (Abbiotec, San Diego, CA, USA) and stimulated with 0.4 mg/mL streptavidin (Sigma, St. Louis, MO, USA) for 30 min at 37 °C. Cells were blocked and stained with PE-conjugated anti-CD63 (BD Biosciences, Mountain View, CA, USA) and propidium iodide to exclude dead cells. Cells were acquired on a FACS Calibur flow cytometer (FACScan; BD Biosciences, Mountain View, CA, USA) and analyzed using FlowJo 7.6 software.

### 4.11. Statistical Data Analysis

After determining the normal distribution of the samples and variance analysis, an unpaired student’s *t*-test was used to determine significant differences (*p*-value) between the two experimental groups. A one-way ANOVA test was used to determine significant differences (*p*-value) between several experimental groups. All results are expressed as mean ± standard error of the mean (SEM).

## 5. Conclusions

Altogether, our data show that MITF and MITF-dependent targets may be therapeutic to target mast cell release abilities and survival in KIT oncogenic mast cell diseases.

## Figures and Tables

**Figure 1 ijms-24-03515-f001:**
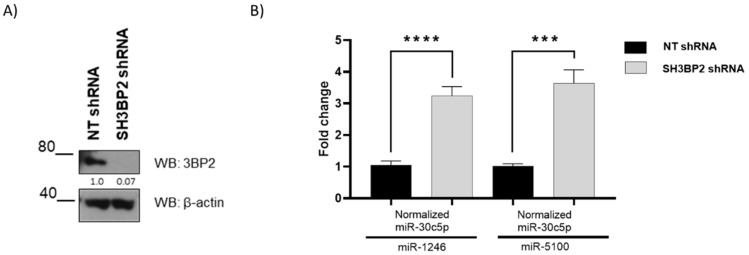
Validation of miR-1246 and miR-5100 upregulation in SH3BP2-silenced HMC-1 cells by real-time PCR. HMC-1 cells were transduced with a non-target shRNA sequence or a specific shRNA SH3BP2. (**A**) A western blot of SH3BP2 (or 3BP2) was performed. (**B**) Validation of miR-1246 and miR-5100 by real-time PCR was carried out; miR-30c5p was used as housekeeping miRNAs. Data are representative of two biological replicates performed in duplicate. Statistical significance (*** *p* < 0.001, **** *p* < 0.0001; Unpaired *t*-test; mean ± SEM).

**Figure 2 ijms-24-03515-f002:**
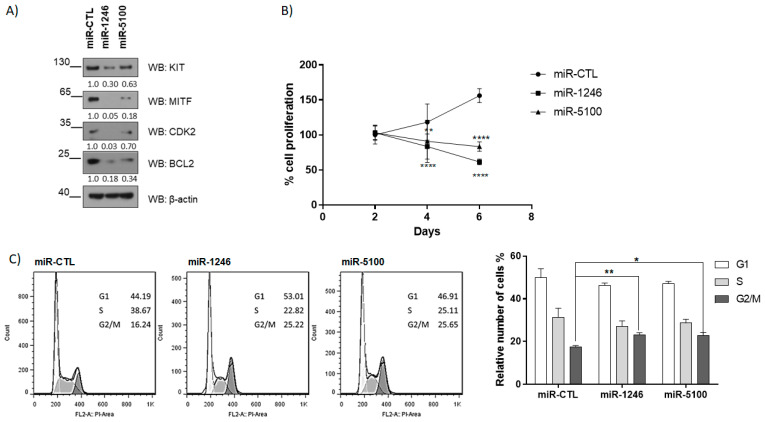
Overexpression of miRNAs affects cell proliferation and cell cycle progression in HMC-1. (**A**) MITF-dependent targets were assessed by western blot at four days post lentiviral transduction. (**B**) WST-1 assay was performed to determine cell proliferation. (** *p* < 0.01, **** *p* < 0.0001; one-way ANOVA; mean ± SEM) n = 3. (**C**) Cell cycle analysis was performed by flow cytometry at six days. Statistical significance (* *p* < 0.05, ** *p* < 0.01); unpaired *t*-test; mean ± SEM n = 3.

**Figure 3 ijms-24-03515-f003:**
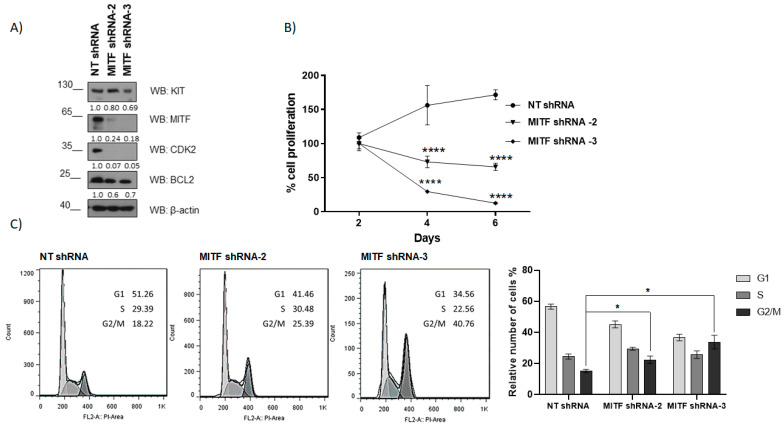
MITF silencing affects cell proliferation and induces cell cycle arrest in HMC-1. (**A**) MITF-dependent targets were assessed by western blot at four days post lentiviral transduction. (**B**) WST-1 assay was performed to determine cell proliferation. **** *p* < 0.0001; one-way ANOVA; mean ± SEM) n = 3. (**C**) Cell cycle analysis by flow cytometry was assessed at six days. Statistical significance (* *p* < 0.05); unpaired *t*-test; mean ± SEM n = 3.

**Figure 4 ijms-24-03515-f004:**
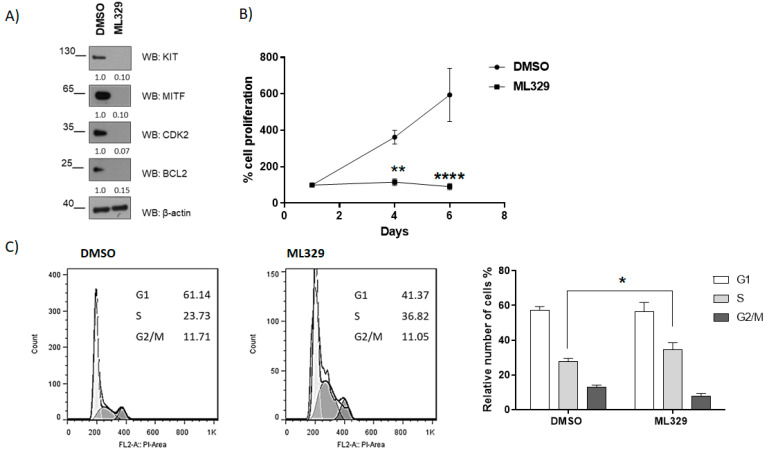
MITF inhibitor impairs cell proliferation and induces cell cycle arrest. (**A**) MITF-dependent targets were assessed by western blot at four days post lentiviral transduction. (**B**) WST-1 assay was performed to determine cell proliferation. (** *p* < 0.01, **** *p* < 0.0001; one-way ANOVA; mean ± SEM) n = 3. (**C**) Cell cycle analysis by flow cytometry was performed at six days. Statistical significance (* *p* < 0.05); unpaired *t*-test; mean ± SEM n = 3.

**Figure 5 ijms-24-03515-f005:**
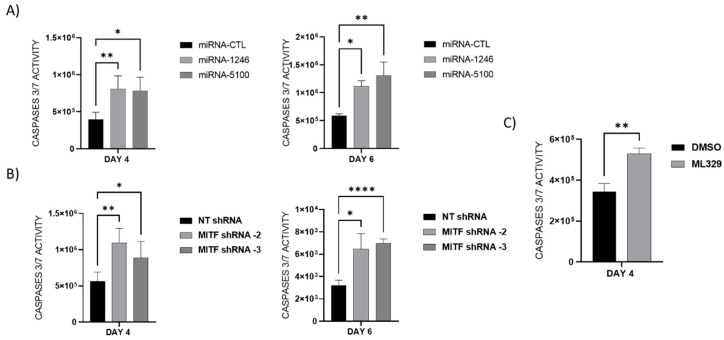
MiR-1246 and miR-5100 overexpression, MITF silencing, and ML329 treatment induce caspases 3/7 activity. Caspase 3/7 activity was measured at 4 and 6 days to examine apoptosis in (**A**) MiR-1246 and miR-5100 overexpression, (**B**) MITF silencing, and (**C**) ML329 treatment. Statistical significance (* *p* < 0.05, ** *p* < 0.01, **** *p* < 0.0001; unpaired *t*-test; mean ± SEM) n = 3.

**Figure 6 ijms-24-03515-f006:**
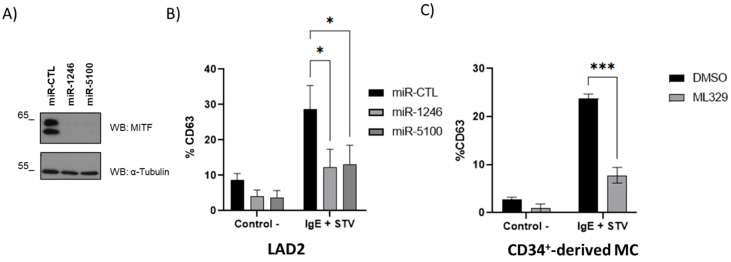
MITF downregulation impairs IgE-dependent degranulation in LAD2- and CD34^+^-derived mast cells. (**A**) Western blot shows MITF levels in miR-1246 and miR-5100 overexpressed LAD2. (**B**) Percentage of CD63 expression in miR-CTL, miR-1246, and miR-5100 overexpressed LAD2 cells. (**C**) Percentage of CD63 expression in CD34^+^-derived mast cells treated with ML329. Data show the mean  ±  SEM. Statistical significance (* *p*  <  0.05, *** *p*  <  0.001; two-way ANOVA with Tukey’s multiple comparison analysis).

## Data Availability

The data that support the findings of this study are available from the corresponding author, M.M., upon reasonable request.

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
