# Peer review of "MITF Downregulation Induces Death in Human Mast Cell Leukemia Cells and Impairs IgE-Dependent Degranulation"

_ijms, 2023, doi:10.3390/ijms24043515_

Round 1
Reviewer 1 Report
In this study, the authors showed that downregulation or inhibition of MITF induces cell cycle arrest and apoptosis of mast cells and also suppresses degranulation, suggesting that MITF may be a therapeutic target for mast cell and KIT-mediated diseases such as allergy and mastcytosis. They also showed that the KIT adaptor molecule SH3BP2 downregulates miR-1246 and miR-5100, which target KIT, and that increased expression of these miRNAs similarly leads to cell cycle arrest and apoptosis of mast cells, as well as suppressing degranulation. The data are clear and interesting. The following concerns should be addressed by the authors. 1. Overexpression of miR-1246 and miR-5100 or downregulation of MITF induced G2/M arrest, while inhibition of MITF induced S phase arrest. What could account for this difference? In addition, the authors described that CDK2 inhibition induces both G1 and G2 arrest, but only G2 arrest was observed in miR-1246 and miR-5100-overexpressing cells and MITF-downregulated cells, although CDK2 expression was significantly reduced in both cells. What about the dependence of cell cycle arrest and apoptosis observed in each experiment on CDK2 and BCL2? Please discuss on these points. 2. The authors showed only representative images from western blotting analysis in Fig. 1A, 2A, 3A, 4A, and 5A. In addition to these images, the results of statistical analysis of band intensities from multiple experiments should be added. 3. How the authors think about the mechanisms underlying the suppression of IgE-mediated degranulation by MITF downregulation or inhibition? 4. The description about the experimental method for GFP-miRNA overexpression is missing. Please add it in the text. 5. Does the KIT gene in HMC-1 cells have some mutations or not? Since it may affect constitutive expression levels of miR-1246 and miR-5100 and MITF, please describe about it.
Author Response
We are grateful for all reviewer's favorable comments on our paper that we believe improve the quality of our manuscript. We have tried to answer all the comments raised by Reviewer #1 as thoroughly as possible.
- Overexpression of miR-1246 and miR-5100 or downregulation of MITF induced G2/M arrest, while inhibition of MITF induced S phase arrest. What could account for this difference? In addition, the authors described that CDK2 inhibition induces both G1 and G2 arrest, but only G2 arrest was observed in miR-1246 and miR-5100-overexpressing cells and MITF-downregulated cells, although CDK2 expression was significantly reduced in both cells. What about the dependence of cell cycle arrest and apoptosis observed in each experiment on CDK2 and BCL2? Please discuss on these points.
Reply: MITF inhibitor ML329 has been proposed to inhibit the MITF pathway and, apart from MITF, may interfere with other molecules. KIT, CDK2, and BCL2 levels were more reduced after ML329 treatment than after miRNAs or shRNAs treatment. CDK2 has been described to regulate both G1/S and G2/M transitions (Hu et al., Mol Cell Biol. 2001 DOI: 10.1128/mcb.21.8.2755-2766.2001 ). Higher expression of double negative CDK2 arrests cells in the mid-S phase. In comparison, lower expression progresses well through an early and mid-S phase but still arrests in late S/G2 (Hu et al., Mol Cell Biol. 2001 DOI: 10.1128/mcb.21.8.2755-2766.2001 ), suggesting that the levels of active CDK2 can be essential in the different phases.
On the other hand, BCL2, an antiapoptotic BCL2 family member, is positively regulated by MITF in imatinib-sensitive and resistant gastrointestinal stromal tumors (Proaño-Perez, E et al. Cancer Gene Therapy. 2022 DOI: 10.1038/s41417-022-00539-1). Moreover, cellular apoptosis triggered by MITF disruption can be rescued by BCL2 overexpression in melanocytes and melanoma (McGill, GG et al. Cell. 2002 DOI: 10.1016/S0092-8674(02)00762-6). Several antiapoptotic proteins have been reported to be MITF-dependent targets. BIRC7, also called ML-IAP (Melanoma inhibitor of apoptosis), which can inhibit extrinsic and intrinsic apoptotic pathways by interaction with caspases, is upregulated by MITF (Hartman and Czyz, J. Investigative Dermatology 2015 DOI: 10.1038/jid.2014.319). The antiapoptotic BCL2A1 has also been identified direct target of MITF, suggesting that the MITF-BCL2A1 relationship may be an intrinsic mechanism protecting melanoma cells from drug-induced death (Haq R et al., PNAs 2013 DOI: 10.1073/pnas.1205575110).
- The authors showed only representative images from western blotting analysis in Fig. 1A, 2A, 3A, 4A, and 5A. In addition to these images, the results of statistical analysis of band intensities from multiple experiments should be added.
Reply: We agree with the reviewer that the image is a representative blot. The reviewer should note that in all the blots where MITF is reduced, MITF-dependent targets, reported in the literature, are also diminished. The data is consistent. The experiments have been performed several times, and some observations have already been published in other cellular models of KIT oncogenic pathologies (GIST cell lines), reinforcing our data in HMC-1.
Fig 1A is an SH3BP2 silencing in mast cells. This silencing in LAD2, HMC-1, and CD34-derived MC has been reported previously by our group (Ainsua-Enrich, e et al. J. Immunology 2012; Ainsua-Enrich, e et al. J. Immunology 2015; Navines-Ferrer et al. Frontiers Immunol, 2019 ). Besides, our group has also reported SH3BP2 silencing in gastrointestinal stromal tumor cells ( Serrano-Candelas, E. et al. Mol. Oncol. 2018; Proaño-Perez, E. et al. Cancers 2022).
Fig 2A, apart from this work, miRNAs 1246 and 1500 silencing MITF and MITF-dependent targets were published recently in GIST cellular models ( Proaño-Perez, E et al., Cancers 2022).
Fig 3A, apart from this work, MITF shRNA reducing CDK2 and BCL2 was published recently in GIST cellular models (Proaño-Perez, Cancer Gene Therapy 2022).
Fig. 4A MITF and MITF-dependent targets reduction give good support for all data.
Fig. 5A is a Caspase experiment performed with luminescence.
We have added supplementary data (Figure S3) with graphs with the results of intensity bands for Figures 2A and 3A, three independent experiments. Note that Figure 4A has been done several times to check MITF levels (however, we have one complete set of MITF and MITF-dependent targets in HMC-1). Nevertheless, the data is very consistent.
- How the authors think about the mechanisms underlying the suppression of IgE-mediated degranulation by MITF downregulation or inhibition?
Reply: In the near future, we are going to pursue this topic and study the underlying mechanism of mast cell degranulation abrogated by MITF knockdown levels.
In that context, MITF-dependent genes are related directly to granules content (Hdc, proteases, chymase) and PGD2 (all are adequately referenced in the manuscript).
On the other hand, MITF function has been related to the biogenesis of lysosomes (Ploper D et al. Proc Natl Acad Sci U S A (2015) doi: 10.1073/pnas.1424576112), and bone marrow-derived MCs from Mitf -/- mice display hypogranularity that can be restored with MITF addition (Shahlaee AH et al. J Immunol (2007) doi: 10.4049/jimmunol.178.1.37852).
Moreover, a recent publication reports that MITF directly regulates STIM1 in melanocytes (Tanwar J. et al. JBC 2022 DOI: 10.1016/J.JBC.2022.102681 ). STIM1 is an endoplasmic reticulum (ER) Ca2+ sensor that, upon activation and decreased ER Ca2+ levels, oligomerizes and activates Orai channels (Orai1/2/3), resulting in Ca2+ influx, a crucial event for granule exocytosis in mast cells (Baba, Y et al. Nature Immunol. 2008. DOI: 10.1038/NI1546).
- The description about the experimental method for GFP-miRNA overexpression is missing. Please add it in the text.
Reply: GFP.miRNA overexpression is carried out by lentiviral technology, as mentioned in the material and methods section in "Lentiviral transduction." Now a new paragraph with complete information has been added.
- Does the KIT gene in HMC-1 cells have some mutations or not? Since it may affect constitutive expression levels of miR-1246 and miR-5100 and MITF, please describe about it.
Reply: HMC-1 cells are characterized to have two KIT mutations, G560V and D860V (the last one is the hallmark of mastocytosis). (Nilsson, G et al. Phenotypic Characterization of the Human Mast‐Cell Line HMC‐1. Scand. J. Immunol. 1994, 39, 489–498, doi:10.1111/j.1365-3083.1994.tb03404).
Indeed, several papers refer to them as a cellular model for mast cell mastocytosis.
Although cells were referenced correctly in material and methods, and D860V mutation was mentioned in the paper, we have added a sentence regarding both mutations in material and methods.
Reviewer 2 Report
In this manuscript the authors describe the role of MITF downregulation in human mast cell leukaemia cells via miRNAs, shRNAs and selective MITF inhibitors. They show that upon downregulation cells proliferate less and go more into apoptosis. As a functional read-out they specifically look at degranulation of mast cells.
The articles gives novel insight into the role of MITF on neoplastic mast cell function. However, the strength of the findings could be improved by using not only one specific method per read-out, especially since the work only relies on in vitro findings. I would suggest to also use additional assays such as AnnexinV / AAD staining or BrdU/EdU assays as further methods and to confirm key findings on proliferation/apoptosis also on LAD2 cells which the authors already have in hand.
Additionally, the work focuses a lot on cellular but not so much on functional aspects of mast cells upon MITF modulation. Could the authors also investigate other aspects such as reduced release of mediators that have been linked to MITF function in MC.
Additionally, MITF was shown to be involved in the anchorage of MC to the BM matrix through regulation of adhesion molecules. Can the authors also show that this is affected/altered upon MITF down-regulation?
Would the authors furthermore expect that the effect of MITF downregulation might be reversed e.g. by activation of external stimuli? This kind of "rescue" experiment could support the mechanistic importance of MITF on MC function.
The proliferation of control transduced cells seems pretty low, since HMC1 cells normally double within 48-72 hours. Normal cell growth is only shown in cells treated with DMSO. Can the authors explain?
Minor:
Which hematologic disorders other than mastocytosis are associated with increased mast cell proliferation?
The sentence about "symptoms exhibited in mastocytosis increase or degrees of mast cell mediators" does not make sense, please rephrase
Were HMC1 or specific clones used in this study?
Figure 1A is not really mentioned/linked in the text.
Supplementary Figure 1A: What is the key message for this figure? Was GFP used as a selection marker? Is this showing the efficacy of the transduction? Please clarify or omit.
Why was b-actin used as a HKG for WB in the first figures, but then the authors changed to a-tubulin? Was there a specific reason?
Author Response
We understand the concerns of reviewer 2. We have tried to answer all the comments raised by Reviewer 2 as thoroughly as possible.
In this manuscript the authors describe the role of MITF downregulation in human mast cell leukaemia cells via miRNAs, shRNAs and selective MITF inhibitors. They show that upon downregulation cells proliferate less and go more into apoptosis. As a functional read-out they specifically look at degranulation of mast cells.
The articles gives novel insight into the role of MITF on neoplastic mast cell function. However, the strength of the findings could be improved by using not only one specific method per read-out, especially since the work only relies on in vitro findings. I would suggest to also use additional assays such as AnnexinV / AAD staining or BrdU/EdU assays as further methods and to confirm key findings on proliferation/apoptosis also on LAD2 cells which the authors already have in hand.Additionally, the work focuses a lot on cellular but not so much on functional aspects of mast cells upon MITF modulation. Could the authors also investigate other aspects such as reduced release of mediators that have been linked to MITF function in MC. Additionally, MITF was shown to be involved in the anchorage of MC to the BM matrix through regulation of adhesion molecules. Can the authors also show that this is affected/altered upon MITF down-regulation?
Reply: These are very interesting questions. We are starting to study the underlying mechanism of IgE-dependent degranulation abrogated by MITF knockdown levels. In the present study, we aimed to search for MITF-dependent prosurvival mechanisms in mast cells harboring oncogènic KIT. Data is consistent, and it has been reported recently in another KIT oncogènic model of gastrointestinal stromal tumors cells by our group (Proaño-Pérez, E. Et al. Cancer Gene Therapy. 2022 ; Proaño-Pérez, E. et al. Cancers 2022).
Our future direction is to address in detail the mediators and signaling involved in the decrease of degranulation. In that context, MITF-dependent genes are related directly to granules content (Hdc, proteases, chymase) and PGD2 (all are adequately referenced in the manuscript). On the other hand, MITF function has been related to the biogenesis of lysosomes (Ploper D et al. Proc Natl Acad Sci U S A (2015) doi: 10.1073/pnas.1424576112), and bone marrow-derived MCs from Mitf -/- mice display hypogranularity that can be restored with MITF addition (Shahlaee AH et al. J Immunol (2007) doi: 10.4049/jimmunol.178.1.37852). Moreover, a recent publication reports that MITF directly regulates STIM1 in melanocytes (Tanwar J. et al. JBC 2022 doi: 10.1016/J.JBC.2022.102681 ). STIM1 is an endoplasmic reticulum (ER) Ca2+ sensor that, upon activation and decreased ER Ca2+ levels, oligomerizes and activates Orai channels (Orai1/2/3), resulting in Ca2+ influx, a crucial event for granule exocytosis in mast cells (Baba, Y et al. Nature Immunol. 2008. doi: 10.1038/NI1546).
Would the authors furthermore expect that the effect of MITF downregulation might be reversed e.g. by activation of external stimuli? This kind of "rescue" experiment could support the mechanistic importance of MITF on MC function.
Reply: MITF levels increase after KIT activation (Lee YN et al. Blood 2006 doi: 10.1182/blood.V108.11.3601.3601). Indeed MITF is downstream of the KIT receptor and governs mast cell differentiation and proliferation (Oppezzo A, et al. Cell Biosci (2021) doi:10.1186/s13578-021-00529-0). KIT receptor signaling regulates MITF through the miR-539 and miR381 downregulation (Lee Y-N, et al. Blood (2011) doi: 10.1182/blood-2010-07-293548). At the same time, MITF fosters KIT expression (Tsujimura T, et al. Blood (1996) 88:1225–1233), showing a reciprocal regulation. In that context, our group reported that the adaptor protein 3BP2 participates in the KIT-MITF axis delivering survival signals in MCs. Thus, silencing of 3BP2 reduced KIT and MITF protein levels and induced MC apoptosis. MITF overexpression rescued KIT protein levels in 3BP2 knockdown cells (Ainsua-Enrich E, et al. J Immunol (2015. doi: 10.4049/jimmunol.1402887(62).
The proliferation of control transduced cells seems pretty low, since HMC1 cells normally double within 48-72 hours. Normal cell growth is only shown in cells treated with DMSO. Can the authors explain?
Reply: Transduced cells are cultured with puromycin to select miRNA or shRNA expression and detect good levels of MITF silencing. The addition of puromycin may explain this reduction in proliferation observed in NTshRNA and miRNA CT (control).
Minor:
Which hematologic disorders other than mastocytosis are associated with increased mast cell proliferation?
Reply: This is a complex matter. Several papers in literature. e.g., from Dr. Peter Valent Diagnosis and classification of mast cell proliferative disorders: delineation from immunologic diseases and non-mast cell hematopoietic neoplasms.JACI. 2004. doi: 10.1016/j.jaci.2004.02.045.
We aim to show MITF's prosurvival role in a KIT oncogènic mast cell model. To my knowledge, HMC-1 is one model that fits this purpose. This cell line harbors the KIT D816V mutation, the hallmark of mastocytosis.
The sentence about "symptoms exhibited in mastocytosis increase or degrees of mast cell mediators" does not make sense, please rephrase
Reply: We appreciate you realized the mistake. It has been corrected in this revised version.
Were HMC1 or specific clones used in this study?
Reply: In this study, we used HMC1 that has two KIT mutations: G560V and D860V (hallmark of mastocytosis). Cells are described in: Nilsson, G.; Blom, T.; Kusche-Gullberg, M.; Kjellen, L.; Butterfield, J.H.; Sundström, C.; Nilsson, K.; Hellman, L. Phenotypic Characterization of the Human Mast‐Cell Line HMC‐1. Scand. J. Immunol. 1994, 39, 489–498, doi:10.1111/j.1365-3083.1994.tb03404.x.
Figure 1A is not really mentioned/linked in the text.
Reply: In the revised version has been indicated in the text.
Supplementary Figure 1A: What is the key message for this figure? Was GFP used as a selection marker? Is this showing the efficacy of the transduction? Please clarify or omit.
Reply: Supplementary Figure 1A shows the efficiency of transduction
Why was b-actin used as a HKG for WB in the first figures, but then the authors changed to a-tubulin? Was there a specific reason?
Reply: In the present study, β-actin was used in all the experiments carried out in HMC-1 cells, while In LAD2 cells, we used αTubulin. Both are good loading controls, and we have used them indistinctly. Lately, we have been performing cellular fraction in LAD2, and we use tubulin to see the purity of cytosolic fraction since actin can also be in the nucleus; honestly, this is the real reason. Nevertheless, the critical fact is to show equal protein levels in all lanes, as shown in the manuscript.
Reviewer 3 Report
This manuscript investigated the role of microphthalmia-associated transcription factor (MITF) in mast cell survival and antigen induced degranulation. SH3 Binding Protein 2 (SH3BP2) downregulation caused miR-1246 and miR-5100 mediated MITF downregulation and mast cell apoptosis and reduced IgE mediated degranulation. Manuscript is well written, and experiments conclusively demonstrated the objectives. However, there are some concerns that need to be addressed
1) If MITF is the downstream of KIT (Line 59-60), then why MITF downregulation diminished KIT expression in Fig 2A?
2) In Fig 3B, why MITF downregulation has very minimal effect on BCL-2? This should be replaced with better blot.
Minor comment:
1) Fig 1 legend is not clear. Authors should rewrite figure legend 1 to show figure 1A is western blot and 1B is RT-PCR.
2) Authors haven’t shown anything on mastocytosis and experiment on HMC-1 cells doesn’t correlate with mastocytosis. Therefore authors shouldn’t stress on mastocytosis in the introduction and discussion.
Author Response
We thank the reviewer for all the comments and suggestions to improve the manuscript. We have tried to answer all the comments raised by Reviewer #3 as thoroughly as possible.
This manuscript investigated the role of microphthalmia-associated transcription factor (MITF) in mast cell survival and antigen induced degranulation. SH3 Binding Protein 2 (SH3BP2) downregulation caused miR-1246 and miR-5100 mediated MITF downregulation and mast cell apoptosis and reduced IgE mediated degranulation. Manuscript is well written, and experiments conclusively demonstrated the objectives. However, there are some concerns that need to be addressed
1) If MITF is the downstream of KIT (Line 59-60), then why MITF downregulation diminished KIT expression in Fig 2A?
Reply: MITF levels increase after KIT activation (Lee YN et al. Blood 2006 doi: 10.1182/blood.V108.11.3601.3601). Indeed MITF is downstream of the KIT receptor and governs mast cell differentiation and proliferation (Oppezzo A, et al. Cell Biosci (2021) doi:10.1186/s13578-021-00529-0). KIT receptor signaling regulates MITF through the miR-539 and miR381 downregulation (Lee Y-N, et al. Blood (2011) doi: 10.1182/blood-2010-07-293548). At the same time, MITF fosters KIT expression (Tsujimura T, et al. Blood (1996) 88:1225–1233), showing a reciprocal regulation. In that context, our group reported that the adaptor protein 3BP2 participates in the KIT-MITF axis delivering survival signals in MCs. Thus, silencing of 3BP2 reduced KIT and MITF protein levels and induced MC apoptosis. MITF overexpression rescued KIT protein levels in 3BP2 knockdown cells (Ainsua-Enrich E, et al. J Immunol (2015. doi: 10.4049/jimmunol.1402887(62).
2) In Fig 3B, why MITF downregulation has very minimal effect on BCL-2? This should be replaced with better blot.
Reply: the reduction of BCL-2 is significant, as shown in the band intensity graph added in Supplementary figure 3. We have better blots just for BCL-2; however, the one presented in the figure is more representative of all proteins analyzed.
Minor comment:
1) Fig 1 legend is not clear. Authors should rewrite figure legend 1 to show figure 1A is western blot and 1B is RT-PCR.
Reply: We have rewritten the figure legend in the revised version.
2) Authors haven't shown anything on mastocytosis and experiment on HMC-1 cells doesn't correlate with mastocytosis. Therefore authors shouldn't stress on mastocytosis in the introduction and discussion.
Reply: We aim to show MITF's prosurvival role in a KIT oncogènic mast cell model. To my knowledge, HMC-1 is one model that fits this purpose. This cell line harbors the KIT D816V mutation, the hallmark of mastocytosis.
Other works have been used for that purpose. To mention some: (Lee YN et al. Blood 2006 doi: 10.1182/blood.V108.11.3601.3601); (Lee Y-N, et al. Blood (2011) doi: 10.1182/blood-2010-07-); (Haenisch B, et al. Immunol Res. 2013 doi: 10.1007/s12026-013-8391-1. )(Arock M, et al. Haematologica . 2018 doi: 10.3324/haematol.2018.195867) Landolina, N et al. Pharmacol. Res 2020.doi 10.1016/j.phrs.2020.104682)
Round 2
Reviewer 1 Report
The authors appropriately responded to most of the comments, but a few points remained to be addressed. 1. Regarding my previous comment 1, the authors responded in the “Author’s Reply“ but did not include a discussion in the text. Please add some discussion in the text. 2. Regarding my previous comment 2, I mistakenly listed Fig 5A when I should have listed Fig 6A.Please correct the legend in Figure S3, where ** and *** are both represented as p<0.01.
Author Response
We want to thank Reviewer one for all the comments and suggestions
The authors appropriately responded to most of the comments, but a few points remained to be addressed. 1. Regarding my previous comment 1, the authors responded in the “Author’s Reply“ but did not include a discussion in the text. Please add some discussion in the text. 2. Regarding my previous comment 2, I mistakenly listed Fig 5A when I should have listed Fig 6A.
Please correct the legend in Figure S3, where ** and *** are both represented as p<0.01.
Reply: We have included some discussion in the new version of the manuscript, regarding previous comment 1 as the reviewer suggested.
Figure S3 has been corrected.
We hope you find this new version suitable for publication
Reviewer 2 Report
I thank the authors for their reply to my comments.
Author Response
We thank reviewer 2 for helpful comments and support
Reviewer 3 Report
Authors properly addressed all my concerns. I recommend this manuscript to be accepted for publication.
Author Response
We thank reviewer comments, suggestions and support